# FLOW MATCHING ON UNORDERED SETS

## ABSTRACT

Flow matching has achieved promising performance across a broad spectrum of data modalities (e.g., image and text). However, there are few works exploring their extension to unordered point sets. Indeed, previous generative models are mostly designed for vector data, with a natural ordering along dimensions. In this paper, we present *unordered flow*, a type of flow-based generative model for generating point sets. Specifically, we propose a lifting approach where we convert unordered data into an appropriate function representation, and learn the probability measure of such representations through function-valued flow matching. For the inverse map from a function representation to unordered data, we introduce a particle filtering method that first warms up the initial particles with Langevin dynamics and then updates them until convergence through gradient-based search. We have conducted extensive experiments on multiple real-world datasets, showing that our *unordered flow* model is highly effective in generating set-structured data and significantly outperforms previous baselines.

## 1 INTRODUCTION

Flow matching (Lipman et al., 2022) and its equivalent: diffusion models (Ho et al., 2020; Song et al., 2021), have achieved promising results in a wide range of data modalities, including images (Dhariwal & Nichol, 2021), videos (Ho et al., 2022), texts (Li et al., 2022), audio (Guan et al., 2024), and tables (Kotelnikov et al., 2023; Jolicoeur-Martineau et al., 2024). However, there are few works exploring their extensions to set-structured data generation, which is essential in many real applications. For example, point cloud generation (Mao et al., 2022), earthquake prediction (Ogata, 1998), and generative view synthesis (Mu et al., 2025).

**Point processes lack flexibility.** In probability theory, unordered data generation are modeled by the point process (Kingman, 1992), with an intensity function to quantify how likely an event occurs. Representative models include the Poisson process (Kingman, 1992) that has constant intensities, and the Hawkes process (Hawkes, 1971) that conditions the intensity on observed events. There are also many works (Mei & Eisner, 2017; Zuo et al., 2020) that parameterized the intensity function with neural networks (e.g., Transformer (Vaswani, 2017)). Despite their simplicity and interpretability, those statistical methods need to explicitly model intensity function, and thus lack parameterization flexibility, resulting in a performance bottleneck. Hence, many subsequent works (Shchur et al., 2020a; Lyu et al., 2022) sought to develop more expressive models.

**Towards unordered generative models.** One such direction is to apply the deep generative models (e.g., GAN (Goodfellow et al., 2020)), which have delivered much better performance than the point process models in generating unordered data. However, generative models are commonly designed for vector data that are naturally ordered in terms of their dimensions, so extra architectural adaptations are required before application. For example, Xie et al. (2021) constructed an energy-based model (LeCun et al., 2006) with a score function that was permutation-invariant to the point set, and Biloš (2021) developed a variant of normalizing flow (Kingma & Dhariwal, 2018) that could model the probability density of unordered sets.

As an advanced generative model, flow matching also needs similar architectural adaptations, but the studies on this topic are quite few at present. Very recently, Lüdke et al. (2023; 2024) proposed to mimic the diffusion process by randomly adding and deleting discrete points. While their method adopted a similar idea of adding noise and denoising, it is still built on top of the point process,

which involves discrete operations (e.g., thinning) and differs from diffusion models (that are based on heat equations (Widder, 1976; Sohl-Dickstein et al., 2015)) in principle. More closely related is Chen & Zhou (2023), which generalized the diffusion process from the continuous space to an integer set, though this in fact assumes that the set-like data has a certain order. It is also worth noting that there are some works (Mu et al., 2025) directly applying diffusion models to unordered data, without considering their inherent permutation invariance.

**Our method: unordered flow.** In this paper, we aim to adapt the flow matching model to unordered set generation, thereby addressing a gap in the literature. To this end, we introduce *unordered flow*, a flow-based generative model that is permutation-invariant to unordered data. Specifically, we represent an unordered set as a type of continuous function that accurately characterizes the discrete structure of the set. The probability measure of this function representation resides in the $L^2$ space (Kantorovich & Akilov, 2014), a nice Hilbert space for function-valued flow matching. We will show that flow matching in the function space can be easily extended from the Euclidean case. For the inverse transform that maps a function representation back to the set, we propose a particle filtering method (Djuric et al., 2003) that randomly initializes a number of particles and performs gradient ascent to iteratively update the particles. Before this gradient-based search, we also explore applying Langevin dynamics (Coffey & Kalmykov, 2012) to warm-up the initial particles, reducing the negative impact of noise.

We have conducted extensive experiments on both synthetic and real-world point sets, across multiple datasets. The results demonstrate that our model can effectively generate unordered data and significantly outperform previous baselines.

## 2 PRELIMINARIES

In this section, we first briefly review the basic of flow-based generative models (Lipman et al., 2022) and then formally define the task of set data generation.

### 2.1 EUCLIDEAN FLOW MATCHING

Let the data dimension be $D_\mathrm{F}$, the essence of flow matching (Lipman et al., 2022) is to interpolate between a noise distribution $p_0 : \mathbb{R}^{D_\mathrm{F}} \to \mathbb{R}^+$ and the potential data distribution $p_1$, leading to a trajectory of marginal distributions $\{p_t\}_{t \in [0,1]}$. This type of interpolation is realized by a temporal-dependent invertible smooth map (i.e., flow) $\boldsymbol{\phi}_t : \mathbb{R}^{D_\mathrm{F}} \to \mathbb{R}^{D_\mathrm{F}}, t \in [0,1]$, satisfying the below ordinary differential equation (ODE):

$$\frac{\mathrm{d}\boldsymbol{\phi}_t(\mathbf{z})}{\mathrm{d}t} = \mathbf{v}_{\theta,t}(\boldsymbol{\phi}_t(\mathbf{z})), \boldsymbol{\phi}_0 = \mathrm{Id}, \tag{1}$$

where vector field $\mathbf{v}_{\theta,t}$ is parameterized by a neural network model and $\mathrm{Id}$ denotes the identity function. The goal is to find a vector field $\mathbf{v}_{\theta,t}$ such that the pushforward $\#$ of flow $\boldsymbol{\phi}_t$ derives the distribution $p_t$: $\#\boldsymbol{\phi}_t[p_0] = p_t$.

The learnable model $\mathbf{v}_{\theta,t}$ is trained through conditional flow matching on observed sample $\mathbf{z}_\mathrm{cond}$. Specifically, a conditional flow is predefined as below:

$$\mathbf{z}_\mathrm{mid} = \boldsymbol{\phi}_t(\mathbf{z} \mid \mathbf{z}_\mathrm{cond}) = (1 - (1 - \zeta)t)\mathbf{z} + t\mathbf{z}_\mathrm{cond}, \tag{2}$$

where $\zeta$ is a small constant, with an analytical form of the conditional vector field as

$$\mathbf{v}_t(\mathbf{z} \mid \mathbf{z}_\mathrm{cond}) = \frac{\mathbf{z}_\mathrm{cond} - (1 - \zeta)\mathbf{z}}{1 - (1 - \zeta)t}. \tag{3}$$

The loss function is the mean square error (MSE) between this vector field and model $\mathbf{v}_{\theta,t}$:

$$\mathcal{J}_\mathrm{E} = \mathbb{E}_{t,\mathbf{z},\mathbf{z}_\mathrm{cond}}[\|\mathbf{v}_{\theta,t}(\mathbf{z}_\mathrm{mid}) - \mathbf{v}_t(\mathbf{z}_\mathrm{mid} \mid \mathbf{z}_\mathrm{cond})\|_2^2], \tag{4}$$

where $t \sim \mathcal{U}\{0,1\}$, $\mathbf{z} \sim p_0$, and $\mathbf{z}_\mathrm{cond} \sim p_1$. Commonly speaking, the initial distribution $p_0$ is pre-determined as a standard Gaussian $\mathcal{G}(\mathbf{0}, \mathbf{I})$.

## 2.2 PROBLEM FORMULATION

An instance $\mathbf{X}$ of unordered data can be defined as a set of vectors in the Euclidean space: $\{\mathbf{x}_i \in \mathbb{R}^{D_{\mathrm{X}}} \mid i \in [1, 2, \cdots, N]\}$. Here $D_{\mathrm{X}}$ denotes the fixed dimension and $N$ is an uncertain integer. Conventionally, unordered data generation is modeled by point processes (Cox & Isham, 1980). In this work, we propose to represent the unordered data $\mathbf{X}$ as some function and aim to model the probability measure of such functions. To be specific, we can uniquely convert any point set $\mathbf{X}$ into a delta representation: $f_{\mathbf{X}} = (\sum_{1 \leq i \leq N} \delta_{\mathbf{x}_i})/N$, where $\delta_{\star}$ denotes the Dirac delta function that centers at point $\star$. All functions of this type form a function space $\mathcal{F}_{\mathrm{delta}}$. To define the probability measure $\mu_{\mathcal{F}_{\mathrm{delta}}}$ on this abstract space, we first denote the underlying probability space that models the randomness of set $\mathbf{X}$ as $(\Omega, \mathcal{B}, \mu_\Omega)$ and the map from a latent sample $\omega \in \Omega$ to the function representation $f_{\mathbf{X}}$ as random variable $F : \Omega \to \mathcal{F}_{\mathrm{delta}}$. Under this scheme, the measure $\mu_{\mathcal{F}_{\mathrm{delta}}}$ can be derived by the pushforward operation: $\#F[\mu_\Omega]$.

In the reminder of this paper, we aim to introduce a variant of flow matching that can model the probability measure $\mu_{\mathcal{F}_{\mathrm{delta}}}$. The main difficulties come from the singularities (e.g., infinity) of delta function $\delta_{\star}$ and non-Euclidean space $\mathcal{F}_{\mathrm{delta}}$.

## 3 METHOD: UNORDERED FLOW

In this part, we present a flow-based generative model: *unordered flow*, for set data generation. We will first address the singularities of function representation $f_{\mathbf{X}}$, and then model its probability measure via function-valued flow matching. Finally, we extract the point set $\widehat{\mathbf{X}}$ from a function $\widehat{f_{\star}}$. Due to the limited space, some implementation details are provided in Appendix D.

### 3.1 RELAXATION OF THE DELTA REPRESENTATION

While the delta representation $f_{\mathbf{X}}$ of unordered set $\mathbf{X}$ has several appealing properties (e.g., simplicity and uniqueness), it is in fact not easy to handle the delta function $\delta_{\star}$ both in practice (e.g., unboundedness) and mathematical analysis (e.g., discontinuity).

**Mixture representation with adaptive variances.** To address the singularities of delta function $\delta_{\star}$, we can relax it with Gaussian approximation: $\mathcal{G}(\star, \epsilon^2 \mathbf{I})$, leading to a continuous representation as $f_{\mathbf{X}, \epsilon} = (\sum_{1 \leq i \leq N} \mathcal{G}(\mathbf{x}_i, \epsilon^2 \mathbf{I}))/N$, where $\epsilon \in \mathbb{R}^+$ is some small constant.

A concern regarding this alternative is that: no matter how small the constant $\epsilon$ is, there might still exist two points $\mathbf{x}_i, \mathbf{y}_j, i \neq j$ whose distance $\|\mathbf{x}_i - \mathbf{y}_j\|_2$ is of the same scale as $\epsilon$. In that case, the Gaussian $\mathcal{G}(\mathbf{x}_i, \epsilon^2 \mathbf{I})$ will fail to approximate delta function $\delta_{\mathbf{x}_i}$, due to its significant overlap with another Gaussian $\mathcal{G}(\mathbf{x}_j, \epsilon^2 \mathbf{I})$. A simple solution to this issue is to adaptively rescale the Gaussian variance in terms of the point-to-point distance as

$$f_{\mathbf{X}, \sigma(\epsilon)} = \frac{1}{N} \sum_{1 \leq i \leq N} \mathcal{G}(\mathbf{x}_i, \sigma_i(\epsilon)^2 \mathbf{I}), \quad \sigma_i(\epsilon) = \epsilon \ln \left( 1 + \min_{j \neq i} \|\mathbf{x}_i - \mathbf{x}_j\|_2 \right). \tag{5}$$

In this way, a point $\mathbf{x}_i$ that is close to another will correspond to a small Gaussian variance, so its approximation $\mathcal{G}(\cdot)$ has a limited impact on other Gaussians. We can also apply an upper clip to the term $\sigma_i(\epsilon)$ in practice, so that the Gaussian at an outlier point will not be too fat.

**Appealing properties.** In the following, we will first see that the mixture representation $f_{\mathbf{X}, \sigma}(\epsilon)$ is sufficient for use if the constant $\epsilon$ is not large.

**Proposition 3.1** (Convergent Representation). *Suppose that the collection of vectors $\mathbf{X}$ is finite, then the limit: $\lim_{\epsilon \to 0} f_{\mathbf{X}, \sigma(\epsilon)} = f_{\mathbf{X}}$, will hold in a weak sense, with a linear convergence speed in terms of the quadratic Wasserstein distance $\mathcal{W}_2$ as*

$$\mathcal{W}_2(f_{\mathbf{X}, \sigma(\epsilon)}, f_{\mathbf{X}}) = \mathcal{O}(\epsilon \sqrt{D_{\mathrm{X}}} \ln \rho), \tag{6}$$

*where $\rho$ is the diameter of set $\mathbf{X}$. Notably, if the set $\mathbf{X}$ is generated by some inhomogeneous Poisson process, with a regular intensity function, then it is finite with probability 1.*

This conclusion not only shows that the limit of representation $f_{\mathbf{X},\sigma(\epsilon)}$ is equal to the delta representation $f_{\mathbf{X}}$, but also verifies that the unordered data $\mathbf{X}$ in our formulation is well-defined: $N < \infty$. Besides, inhomogeneous Poisson processes can generalize many types of unordered data, and thus our assumption is not strong. For this reason, Kidger et al. (2020) adopted it to parameterize neural ODE. The proof for this conclusion is provided in Appendix A.

We define the resulting function space that consists of all possible mixture representation $f_{\mathbf{X},\sigma}(\epsilon)$ as $\mathcal{F}_{\mathrm{mix}}$, with a probability measure denoted as $\mu_{\mathcal{F}_{\mathrm{mix}}}$.

**Proposition 3.2** (Regular Function Space). *The space $\mathcal{F}_{\mathrm{mix}}$ of mixture representation $f_{\mathbf{X},\sigma(\epsilon)}$ is contained in the space of square-integrable functions: $\mathcal{L}^2(\mathbb{R}^{D_{\mathrm{X}}}) \supseteq \mathcal{F}_{\mathrm{mix}}$. In this sense, an immediate corollary is that the probability measure $\mu_{\mathcal{F}_{\mathrm{mix}}}$ is supported on the $L^2$ space. However, this is not the case for the probability measure $\mu_{\mathcal{F}_{\mathrm{delta}}}$ of delta representation $f_{\mathbf{X}}$.*

This claim is significant in that the new measure $\mu_{\mathcal{F}_{\mathrm{mix}}}$ resides in a nice function space: $L^2$. For example, this space can accommodate the Gaussian measure (i.e., infinite-dimensional Gaussian) and has a naturally defined inner product: $\langle g, h \rangle_{\mathrm{L2}} = \int g(\mathbf{y})h(\mathbf{y})d\mathbf{y}$, which facilitates mathematical analysis. The proof for this conclusion is provided in Appendix B.

## 3.2 FUNCTION-VALUED GENERATIVE FLOW

Developing a generative model for the mixture representation $f_{\mathbf{X},\sigma(\epsilon)}$ seems not trivial, as it is a infinite-dimensional function. Despite some minor differences in notation, both flow-based and diffusion-based generative models can be easily extended to the function setting. The foundation of such extensions was rigorously built by many previous works (Kuo, 2006; Williams & Rasmussen, 2006; Lim et al., 2023; Pidstrigach et al., 2023; Kerrigan et al., 2024).

**Generalized flow on the $L^2$ function space.** The $L^2$ space $\mathcal{L}^2(\mathbb{R}^{D_{\mathrm{X}}})$, which supports the probability measure $\mu_{\mathcal{F}_{\mathrm{mix}}}$ (as indicated in Proposition 3.2), can be paired with inner product $\langle \cdot, \cdot \rangle_{\mathrm{L2}}$ to form a Hilbert space $\mathcal{H}_{\mathrm{L2}}$. Similar to the Euclidean case (i.e., Sec. 2.1), the core of flow matching in the space $\mathcal{H}_{\mathrm{L2}}$ is a function-valued vector field $\mathbf{u}_{\theta,t} : \mathcal{H}_{\mathrm{L2}} \to \mathcal{H}_{\mathrm{L2}}, t \in [0,1]$ parameterized by neural networks (Li et al., 2021). Through an "ODE" that has a very similar form in the Euclidean case (i.e., Eq. (1)), this vector field $\mathbf{u}_{\theta,t}$ can be used to generate the flow as

$$\frac{\mathrm{d}}{\mathrm{d}t}[\boldsymbol{\varphi}_t(h)] = \mathbf{u}_{\theta,t}(\boldsymbol{\varphi}_t(h)), \boldsymbol{\varphi}_0 = \mathrm{Id}, \tag{7}$$

where $\mathrm{d}/\mathrm{d}t[\cdot]$, $\mathrm{Id}$ are respectively the generalized differential (Agarwal & O'Regan, 1998) and identity operators on the Hilbert space. As expected, the pushforward $\#$ of this flow $\boldsymbol{\varphi}_t$ interpolates between a Gaussian measure $\eta_0 = \mathcal{G}(\mathbf{0}, \boldsymbol{\Gamma})$ at the time step $t = 0$ and the target measure $\eta_1 = \mu_{\mathcal{F}_{\mathrm{mix}}}$ at step $t = 1$: $\eta_t = \#\boldsymbol{\varphi}_t[\eta_0]$. A difference is that the covariance coefficients of Gaussian $\mathcal{G}$ is a well-defined (e.g., symmetric) linear operator $\boldsymbol{\Gamma} : \mathcal{H}_{\mathrm{L2}} \to \mathcal{H}_{\mathrm{L2}}$.

**Training with conditional flow.** Following the Euclidean case (i.e., Eq. (2)), the training of model $\mathbf{u}_{\theta,t}$ also relies on a conditional flow, which is predefined as

$$h_{\mathrm{mid}} = \boldsymbol{\varphi}_t(h \mid h_{\mathrm{cond}}) = (1 - (1 - \zeta)t)h + th_{\mathrm{cond}}, \tag{8}$$

with a closed-form conditional vector field:

$$\mathbf{u}_t(h \mid h_{\mathrm{cond}}) = \frac{h_{\mathrm{cond}} - (1 - \zeta)h}{1 - (1 - \zeta)t}. \tag{9}$$

Lastly, analogous to Euclidean case $\mathcal{J}_{\mathrm{E}}$ (i.e., Eq. (4)), a MSE-like loss can be derived as

$$\mathcal{J}_{\mathrm{H}} = \mathbb{E}[\|\mathbf{u}_{\theta,t}(h_{\mathrm{mid}}) - \mathbf{u}_t(h_{\mathrm{mid}} \mid h_{\mathrm{cond}})\|_{\mathrm{L2}}^2], \tag{10}$$

where the expectation $\mathbb{E}$ is taken over $t \sim \mathcal{U}\{0,1\}, h \in \eta_0, h_{\mathrm{cond}} \in \eta_1$ and the squared norm $\| \star \|_{\mathrm{L2}}^2$ is induced by the inner product as $\langle \star, \star \rangle_{\mathrm{L2}}$.

## 3.3 GRADIENT-BASED INVERSE TRANSFORM

While we can generate mixture representation $\widehat{f}_{\star,\sigma(\epsilon)}$ from a trained *unordered flow* model $\mathbf{u}_{\theta,t}$, it is not obvious how to extract the potential unordered data $\widehat{\mathbf{X}} \subset \mathbb{R}^{D_{\mathrm{X}}}$ hiding in the input domain of this continuous function: $\widehat{f}_{\star,\sigma(\epsilon)} : \mathbb{R}^{D_{\mathrm{X}}} \to \mathbb{R}$.

---

**Algorithm 1:** Inverse Transform: $\widehat{f}_{\star,\sigma(\epsilon)} \mapsto \widehat{\mathbf{X}}$

---

**Input:** Function representation $\widehat{f}_{\star,\sigma(\epsilon)}$ generated by the model $\mathbf{u}_{\theta,t}$
**Output:** Unordered set of vectors: $\widehat{\mathbf{X}}$
Randomly sample a set of particles $\mathbf{Y}^{(1)}$ from region $\mathcal{X}$
**for** $s = 1, 2, \cdots S_{\text{lgvin}} - 1$ **do**
$\quad$ Warm-up with Langevin dynamics: $\mathbf{Y}^{(s)} \mapsto \mathbf{Y}^{(s+1)}$, in terms of Eq. (12)
**for** $s = S_{\text{lgvin}}, S_{\text{lgvin}} + 1, \cdots, S_{\text{lgvin}} + S_{\text{grad}} - 1$ **do**
$\quad$ Search with gradient ascent: $\mathbf{Y}^{(s)} \mapsto \mathbf{Y}^{(s+1)}$, in terms of Eq. (11)
Merge converged particles: $\widehat{\mathbf{Y}} = \mathbf{Y}^{(S_{\text{lgvin}}+S_{\text{grad}})} \mapsto \widehat{\mathbf{X}}$

---

**Gradient-based search.** If the constant $\epsilon$ is sufficiently small, then the unordered set $\widehat{\mathbf{X}}$ can be approximately regarded as the set of local maxima of function $\widehat{f}_{\star,\sigma(\epsilon)}$. Therefore, a naive idea is to first start from a random point $\mathbf{y} \in \mathbb{R}^{D_{\mathbf{X}}}$ and then perform gradient ascent to iteratively update it until converging to some point in set $\widehat{\mathbf{X}}$. Specifically, suppose that the set $\widehat{\mathbf{X}}$ is contained in a bounded region $\mathcal{X} \subset \mathbb{R}^{D_{\mathbf{X}}}$, we sample a number of initial particles $\mathbf{Y}^{(1)} = \{\mathbf{y}_i^{(1)}\}_{1 \le i \le M}, M \in \mathbb{N}^+$ from the region $\mathcal{X}$ and iterate particle updates as

$$\mathbf{Y}^{(s+1)} = \{\mathbf{y}_i^{(s+1)} \mid \mathbf{y}_i^{(s)} \in \mathbf{Y}^{(s)}\}, \quad \mathbf{y}_i^{(s+1)} = \mathbf{y}_i^{(s)} + \alpha \nabla \widehat{f}_{\star,\sigma(\epsilon)}(\mathbf{y}_i^{(s)}), \tag{11}$$

where superscript $s$ is counted $S_{\text{grad}} \in \mathbb{N}^+$ times and step size $\alpha$ is a small number. Empirically, we find that this gradient-based method is very effective, except that it might not work well when the constant $\epsilon$ is improperly large. In that case, there might exist some *noisy peaks*, which are still a local maximum of function $f_{\mathbf{X},\sigma(\epsilon)}$ but do not belong to the point set $\mathbf{X}$, misguiding the gradient-based search process. For example, the function frequently fluctuates in a certain region, resulting in many "misleading" local maximum. The same problem might also occur when the model $\mathbf{u}_{\theta,t}$ is not well trained, generating a representation $\widehat{f}_{\star,\sigma(\epsilon)}$ that are not mixture-like.

**Langevin warm-up.** To handle *noisy peaks*, a key perspective is that the mixture representation $f_{\mathbf{X},\sigma(\epsilon)}$ actually forms a certain density function:

$$\int f_{\mathbf{X},\sigma(\epsilon)}(\mathbf{y})d\mathbf{y} = \int \Big( \sum_{1 \le i \le N} \frac{1}{N} \mathcal{G}(\mathbf{y}; \mathbf{x}_i, \sigma_i(\epsilon)^2 \mathbf{I}) \Big) d\mathbf{y} = \frac{1}{N} \sum_{1 \le i \le N} \Big( \int \mathcal{G}(\mathbf{y}; \cdot) d\mathbf{y} \Big) = 1,$$

and those undesired peaks are with much lower densities than the points in set $\mathbf{X}$. Therefore, we propose to apply Langevin dynamics (Coffey & Kalmykov, 2012) to warm up the uniformly initialized points $\mathbf{Y}^{(1)}$ before performing gradient-based search (i.e., Eq. (11)):

$$\mathbf{y}_i^{(s+1)} = \mathbf{y}_i^{(s)} + \beta \nabla \ln \widehat{f}_{\star,\sigma(\epsilon)}(\mathbf{y}_i^{(s)}) + \sqrt{2\beta} \mathbf{z}_i^{(s)}, \tag{12}$$

where $i \in [1, M]$ and $\mathbf{z}_i^{(s)} \sim \mathcal{G}(\mathbf{1}, \mathbf{I})$ is a standard Gaussian noise. The use of this warm-up is to attract random points $\mathbf{Y}^{(1)}$ to high density areas. We have explained why Langevin dynamics can achieve this goal in Appendix C. Simply put, Langevin dynamics is a type of Markov chain Monte Carlo (MCMC) (Andrieu et al., 2003), which can converge into a desired distribution.

During *Langevin warm-up*, the superscript $s$ counts from 1 to a given number $S_{\text{lgvin}} \in \mathbb{N}^+$, and after that, it continues to increase to $S_{\text{lgvin}} + S_{\text{grad}}$ in the gradient-based search process. We denote the final particle set $\mathbf{Y}^{(S_{\text{lgvin}}+S_{\text{grad}})}$ as $\widehat{\mathbf{Y}} = \{\widehat{\mathbf{y}}_i\}_{1 \le i \le M}$ for convenience.

**De-duplication as noise filtering.** The number of particles $M$ should be relatively large so that the search result $\widehat{\mathbf{Y}}$ can fully cover the potential point set $\widehat{\mathbf{X}}$, though this will cause redundancy: different particles converge to the same point. On the other hand, considering that the search result $\widehat{\mathbf{Y}}$ might still contain noises, such redundancy is useful since it indicates how unlikely a particle is noisy, based on whether the particle overlaps with a enough number of other particles.

In this spirit, a simple but very effective de-duplication procedure is designed as follows:

1. Merge close particles in the search result $\widehat{\mathbf{Y}}$ via single-pass clustering (Papka et al., 1998; Behnezhad et al., 2023), resulting in a collection of disjoint particle groups;

2. Filter out the groups that are small in size, since they are not reliable;

3. Take an average over each group, and all such numbers form the set $\widehat{\mathbf{X}}$.

The clustering method in the first step is very simple. Specifically, we pick up one particle at a time, adding it to the closest group within a predefined range; If existing groups are all distant, we initiate a new group that includes the particle.

## 4 RELATED WORK

There are a number of papers in the literature focusing on unordered data generation. We classify these papers into three categories, and respectively discuss them as follows.

**Classical methods based on point processes.** Conventionally in probability theory, point set generation are modeled by the point process (Cox & Isham, 1980). The core of that framework is the intensity function, indicating how likely an event will occur. For example, the famous Poisson process (Kingman, 1992) corresponds to a constant intensity function, while its inhomogeneous version means that the function can be arbitrarily valued. More complex models include the Cox process (Cox, 1955) that sets the intensity function to be stochastic and the Hawkes process (Hawkes, 1971) that conditions the intensity function on observed events to capture cross-event dependencies (e.g., self-excitation). The intensity function can be parameterized by neural networks (Omi et al., 2019) or kernel mixture (Okawa et al., 2019). For example, Mei & Eisner (2017) applied a variant of LSTM (Graves & Graves, 2012) to parameterize the Hawkes processes, while Zhang et al. (2020); Zuo et al. (2020) chose Transformer (Vaswani, 2017). In the special case of temporal point processes, the data can in fact be ordered in terms of the timestamp. Therefore, some previous works (Du et al., 2016; Shchur et al., 2020a) directly learned the time interval distribution, without modeling the intensity function.

For this class of methods, they are either of limited expressiveness (e.g., Hawkes Process) or suppose that the set-like data still have some temporal dimension (i.e., not truly unordered) (Zuo et al., 2020), which is less general than our proposed *unordered flow*.

**Methods based on deep generative models** As deep generative models (e.g., GAN (Goodfellow et al., 2020)) have gained great popularity, there is a surge of interest adapting those techniques to set data generation (Yang et al., 2019; Kim et al., 2021; Yang et al., 2022). For example, Biloš (2021) learned a permutation-invariant distribution via normalizing flow (Kingma & Dhariwal, 2018), and Xie et al. (2021) designed a type of energy-based model (LeCun et al., 2006) for unordered point cloud. With the emergence of diffusion-based generative model, there are also some explorations (Luo & Hu, 2021; Lyu et al., 2022) that directly applied diffusion models to unordered data (e.g., spatial coordinates), without considering their permutation invariance.

When the temporal dimension exists, some works (Kidger et al., 2020; Chen et al., 2021; Yuan et al., 2023) adopted neural ODE (Chen et al., 2018) or Transformer to model the irregularly sampled events, with diffusion models or normalizing flow learning the spatial distribution.

**Very recent diffusion-like baselines** Until recently, there are a few attempts that introduce a special type of generative models to unordered data, which mimic how diffusion models work: adding noise and denoising. For example, Lüdke et al. (2023; 2024) reformulated the diffusion process as randomly excluding original points and adding noisy ones. This type of methods in fact differ from the conventional diffusion models (which are based on heat equations (Sohl-Dickstein et al., 2015)) in principle, involving many discrete operations (e.g., thinning).

Closely related to this paper, Chen & Zhou (2023) introduced JUMP that generalizes the diffusion process from the continuous space to an integer set, and Biloš et al. (2023) proposed to regard irregular time series as a continuous function for generative modeling. However, those baselines in fact assume that the data exhibits a certain order (e.g., temporal dimension), which are inapplicable to truly unordered data (e.g., point cloud).

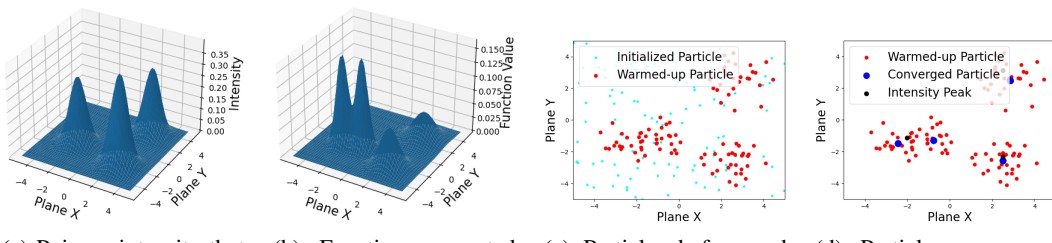

(a) Poisson intensity that generates training data.

(b) Function generated by *unordered flow*.

(c) Particles before and after warm-up.

(d) Particles converged via gradient ascent.

Figure 1: Our *unordered flow* model $\mathbf{u}_{\theta,t}$ applied to an inhomogeneous Poisson process. The left two subfigures illustrates the Poisson intensity function and a mixture representation $\widehat{f}_{\star,\sigma(\epsilon)}$ generated from the model. The right two respectively depict the warm-up of uniformly initialized particles $\mathbf{Y}^{(1)}$ and their convergence to the point set $\widehat{\mathbf{X}}$.

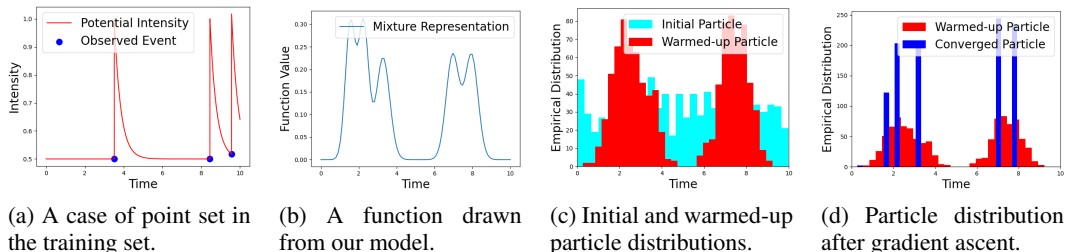

(a) A case of point set in the training set.

(b) A function drawn from our model.

(c) Initial and warmed-up particle distributions.

(d) Particle distribution after gradient ascent.

Figure 2: Results from our *unordered flow* model $\mathbf{u}_{\theta,t}$ on the Hawkes process. The left two subfigures illustrate a point set $\mathbf{X}$ in the training set and a mixture representation $\widehat{f}_{\star,\sigma(\epsilon)}$ sampled from our model. The right two show the empirical distributions of particles, including their initial $\mathbf{Y}^{(1)}$, warmed-up $\mathbf{Y}^{(S_{\text{lgvin}})}$, and converged versions $\widehat{\mathbf{Y}}$.

## 5 EXPERIMENTS

In this section, we aim to verify the effectiveness of our *unordered flow* model. We show that it performs well on both synthetic and real datasets, outperforming previous baselines. Other results (e.g., ablation studies) also verify that every part of our model is useful. More experiment results and minor setup details are respectively provided in Appendix G and Appendix F.

**Benchmark datasets**  We adopt both synthetic and real-world datasets for making comparison. Specifically, we synthesize training data using two common point processes: one is a collection of point sets sampled from a 2D inhomogeneous Poisson process, with its intensity set as a squared-exponential mixture, while the other are drawn from the 1D Hawkes process, with an exponential kernel parameterizing its intensity function.

For the real datasets, we follow previous works (Chen et al., 2018; Lüdke et al., 2024) to adopt three real-world datasets of point sets: Japan Earthquakes (U.S. Geological Survey, 2020), New Jersey COVID-19 Cases (The New York Times, 2020), and Citibike Pickups (Citi Bike, 2024). The pre-processing and splits of these datasets are consistent with Chen et al. (2021).

**Evaluation metrics**  Following many previous works (Shchur et al., 2020b; Lüdke et al., 2023; 2024), we adopt two evaluation metrics: 1) *S-WStein*, the Wasserstein distance (Ramdas et al., 2017) that measures the distributional discrepancy between the size of generated point sets and that of real sets; 2) *D-MMD*, the maximum mean discrepancy measure (MMD) (Gretton et al., 2012) that quantifies the distribution gap between two point processes. We did not include negative log-likelihood (NLL) because many types of generation models (e.g., GAN) are not a density estimator and there are many studies (Theis et al., 2016; Shchur et al., 2020b) pointing out that it is in fact not a proper metric. The lower both metrics are, the better the model performs.

| Method | Earthquakes | | COVID-19 | | Citibike | |
|---|---|---|---|---|---|---|
| | S-WStein | D-MMD | S-WStein | D-MMD | S-WStein | D-MMD |
| Log-Gaussian Cox Process | 0.047 | 0.214 | 0.209 | 0.340 | 0.104 | 0.336 |
| Permutation-invariant Normalizing Flow | 0.043 | 0.191 | 0.185 | 0.271 | 0.071 | 0.105 |
| Energy-based Generative PointNet | 0.040 | 0.185 | 0.201 | 0.253 | 0.063 | 0.091 |
| Point Set Diffusion | 0.038 | 0.173 | 0.199 | 0.268 | 0.056 | 0.092 |
| Our Model: *Unordered Flow* | **0.023** | **0.125** | **0.153** | **0.213** | **0.041** | **0.079** |

Table 1: The performances of our model and baselines on three real-world datasets. The two metrics: *S-WStein* and *D-MMD*, respectively measure the distributional discrepancies between the generated and real point sets in terms of the set size and point position. Plus, the results from our implemented models are averaged over 10 runs.

| Method | Earthquakes | | COVID-19 | |
|---|---|---|---|---|
| | S-WStein | D-MMD | S-WStein | D-MMD |
| Our Model: *Unordered Flow* | **0.023** | **0.125** | **0.153** | **0.213** |
| w/o Adaptive Variance (i.e., Eq. (5)) | 0.028 | 0.136 | 0.185 | 0.232 |
| w/o *Langevin Warm-up* (i.e., Eq. (12)) | 0.031 | 0.146 | 0.176 | 0.239 |
| w/o Noisy Peak Filtering | 0.033 | 0.143 | 0.181 | 0.229 |

Table 2: Our ablation studies. Each of the last three rows shows the degraded model performance after excluding a specific module (e.g., *Langevin warm-up*) from *unordered flow*.

**Baselines**  We adopt multiple key baselines for comparison, including log-Gaussian Cox process (Møller et al., 1998): a known point process model, Generative PointNet (Xie et al., 2021) and order-insensitive Normalizing Flow (Biloš, 2021): deep generative models extended to unordered data, and Point Set Diffusion (Lüdke et al., 2024): a diffusion-like generative models but with a different working principle from typical diffusion models. Each adopted baseline has been comprehensively discussed in Sec. 4. The results of Cox process and Point Set Diffusion are copied from Lüdke et al. (2024), and those of other two baselines come from our implementation.

### 5.1 PROOF-OF-CONCEPT STUDIES

We aim to first verify whether our *unordered flow* model can approximate two common point processes: inhomogeneous Poisson process and Hawkes process, with an emphasize to demonstrate how its workflow runs in practice.

The results on the Poisson process are shown in Fig. 1. From Subfig. 1b, we can see that our model generates a function $\widehat{f}_{\star,\sigma(\epsilon)}$ that is consistent with the form of mixture representation $f_{\mathbf{X},\epsilon}$ in the training data, with density peaks located in the high-intensity areas shown in Subfig. 1a. As introduced in Sec. 3.3, to decode the potential point set $\widehat{\mathbf{X}}$ from the sampled function $\widehat{f}_{\star,\sigma(\epsilon)}$, we first uniformly sample a number of particles $\mathbf{Y}^{(1)}$, with *Langevin warm-up* moving them towards high-density regions. This warm-up process is illustrated in Subfig. 1c, which successfully achieves our goal. Finally, Subfig. 1d shows that the warmed-up particles $\mathbf{Y}^{(S_{\mathrm{lgvin}})}$ perfectly converge to the density peaks of function $f_{\star,\epsilon}$ (i.e., hidden point set $\widehat{\mathbf{X}}$) after gradient-based search.

The results for the Hawkes process in Fig. 2 are also promising. Similarly, the generated function from our *unordered flow* is mixture-like (Subfig. 2b), and the initial particles are warmed-up well (Subfig. 2c), with convergence to the density peaks of the sampled function (Subfig. 2d). The results on both Poisson and Hawkes processes verify that every module of *unordered flow* is effective, and all these modules form a well-performing generative model for point processes.

### 5.2 EVALUATION ON REAL DATASETS

We also applied *unordered flow* to real datasets, evaluating its performance in practice. The results are shown in Table 1. We can see that our model significantly outperforms previous baselines on all the datasets and in terms of each metric. For example, our model has achieved a much lower *D-MMD* score than the latest baseline (i.e., Point Set Diffusion) by 20.52% on the COVID-19 dataset, which

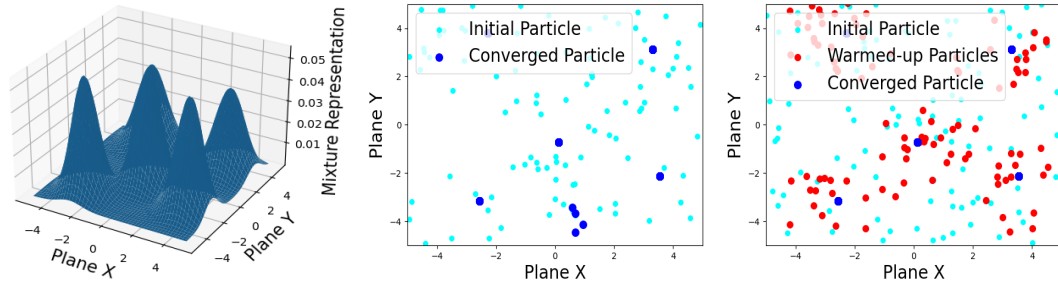

(a) A function sampled from our *unordered flow* model.

(b) Initial particles converge but some fall into "noisy peaks".

(c) Initial particles are warmed-up and thus converge efficiently.

Figure 3: A case study to show the inverse transform, which aims to identifies the point set $\widehat{\mathbf{X}}$ from a mixture representation $\widehat{f}_{\star,\sigma(\epsilon)}$ (i.e., left subfigure). The middle subfigure is the convergence of particles without *Langevin warm-up*, while the right one is with the warm-up.

is $39.47\%$ for *S-WStein* on the Earthquakes dataset. These notably improvements over baselines strongly confirm the effectiveness of *unordered flow*.

An interesting observation from the table is that a previous energy-based generative baseline (i.e., Generative PointNet) performed competitively with the very recent Point Set Diffusion, which is still built on top of the framework of point processes. For example, their performance gap on the Citibik dataset is only $1.09\%$ in terms of *D-MMD*. This indicates that the direction of applying deep generative models to unordered data is promising.

### 5.3    ABLATION STUDIES

We have conducted ablation experiments on two real datasets to verify the effectiveness of multiple modules in our *unordered flow* model. The results are shown in Table 2. We can see that every tested module is crucial to our model performance. For example, the last row in the table means not to exclude small groups after clustering converged particles. This leads to a significant increase by $18.30\%$ on the COVID-19 dataset in terms of *S-WStein*.

### 5.4    CASE STUDY

While the ablation studies have quantitatively verified that *Langevin warm-up* is essential to the model performance, we showcase this point by a real example.

The case is demonstrated in Fig. 3. From Subfig. 3b, we can see that the uniformly initialized particles $\mathbf{Y}^{(1)}$ without *Langevin warm-up* might converge to some "noisy peaks" (mentioned in Sec. 3.3) in the bottom center, and this problem can be perfectly addressed by the warm-up trick as shown in Subfig. 3c. Another observation is that *Langevin warm-up* greatly speeds up the particle filtering process. The convergence in Subfig. 3c only takes $\mathcal{S}_{\mathrm{lgvin}} + S_{\mathrm{grad}} = 500$ iterations in total, which is 5000 in Subfig. 3b.

## 6    CONCLUSION

In this paper, we present *unordered flow*, a type of flow-based generative model for unordered data generation, filling a gap in the literature. The core of this technique is to map the unordered data into a mixture representation in the function space, which is accurate and derives a nice probability measure supported on the $L^2$ space, facilitating function-valued flow matching. For the inverse transform that maps a given function back to the unordered data, we propose a particle filtering method, with Langevin dynamics to warm up the particles and gradient ascent to update them until convergence. Extensive experiments on both synthetic and real datasets show that our model can generate set data well, significantly outperforming previous key baselines.

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

# A    PROOF: CONVERGENCE OF THE MIXTURE REPRESENTATION

We will first prove the first and last claims, and then go to the second one.

## A.1    THE LIMIT WITH A REDUCIBLE ASSUMPTION

Since the point set $\mathbf{X}$ is finite (i.e., $N < \infty$), we can infer that there exists a finite number $\rho$ satisfying

$$\infty > \rho > \sup_{i \neq j} \|\mathbf{x}_i - \mathbf{x}_j\|_2. \tag{13}$$

A known fact is that the Gaussian $\mathcal{G}(\mathbf{x}, \epsilon^2 \mathbf{I})$ weakly converges to delta function $\delta_{\mathbf{x}}$ as $\epsilon \to 0$. Specifically, for any smooth test function $g$, this means: given any $\epsilon_2 > 0$, there always exists a number $\epsilon_1 > 0$ such that

$$\left| \int \mathcal{G}(\mathbf{y}; \mathbf{x}, \epsilon^2 \mathbf{I}) g(\mathbf{y}) d\mathbf{y} - \int \delta_{\mathbf{x}}(\mathbf{y}) g(\mathbf{y}) d\mathbf{y} \right| < \epsilon_2 \tag{14}$$

holds for any $\epsilon \in (0, \epsilon_1]$. In terms of this fact, given the error $\epsilon_2$, there always exists a finite set of numbers $\{\epsilon_i\}_{i \in [1,N]}$ such that the below inequality:

$$\left| \int (\mathcal{G}(\mathbf{y}; \mathbf{x}_i, \epsilon^2 \rho^2 \mathbf{I}) - \delta_{\mathbf{x}_i}(\mathbf{y})) g(\mathbf{y}) d\mathbf{y} \right| < \epsilon_2, \tag{15}$$

holds for any $i \in [1, N], \epsilon < (0, \epsilon_1]$. Following these inequalities, we have

$$\left| \int (f_{\mathbf{X},\sigma(\epsilon)}(\mathbf{y}) - f_{\mathbf{X}}(\mathbf{y})) g(\mathbf{y}) d\mathbf{y} \right|$$

$$= \left| \int \left( \sum_{1 \leq i \leq N} \frac{1}{N} \mathcal{G}(\mathbf{y}; \mathbf{x}_i, \sigma_i(\epsilon)^2 \mathbf{I}) - \sum_{1 \leq i \leq N} \frac{1}{N} \delta_{\mathbf{x}_i}(\mathbf{y}) \right) g(\mathbf{y}) d\mathbf{y} \right|$$

$$= \left| \frac{1}{N} \sum_{1 \leq i \leq N} \int (\mathcal{G}(\mathbf{y}; \mathbf{x}_i, \sigma_i(\epsilon)^2 \mathbf{I}) - \delta_{\mathbf{x}_i}(\mathbf{y})) g(\mathbf{y}) d\mathbf{y} \right| \tag{16}$$

$$< \frac{1}{N} \sum_{1 \leq i \leq N} \left| \int (\mathcal{G}(\mathbf{y}; \mathbf{x}_i, \sigma_i(\epsilon)^2 \mathbf{I}) - \delta_{\mathbf{x}_i}(\mathbf{y})) g(\mathbf{y}) d\mathbf{y} \right| < \frac{1}{N} \sum_{1 \leq i \leq N} \epsilon_2 = \epsilon_2,$$

where the last inequality holds because

$$\sigma_i(\epsilon) = \epsilon \ln \left( 1 + \min_{j \neq i} \|\mathbf{x}_i - \mathbf{x}_j\|_2 \right) \leq \epsilon \min_{j \neq i} \|\mathbf{x}_i - \mathbf{x}_j\|_2 \leq \epsilon \left( \sup_{j \neq k} \|\mathbf{x}_j - \mathbf{x}_k\|_2 \right) = \epsilon \rho. \tag{17}$$

Therefore, for any smooth test function $g$, we can infer that

$$\lim_{\epsilon \to 0} \int f_{\mathbf{X},\sigma(\epsilon)}(\mathbf{y}) g(\mathbf{y}) d\mathbf{y} = \int f_{\mathbf{X}}(\mathbf{y}) g(\mathbf{y}) d\mathbf{y}. \tag{18}$$

With this result, it is known in the distribution theory (Friedlander & Joshi, 1998) that the mixture representation $f_{\mathbf{X},\sigma(\epsilon)}(\mathbf{y})$ is equal to the delta representation $f_{\mathbf{X}}(\mathbf{y})$ in a weak sense.

For the second claim, the intensity function $\lambda(\mathbf{y}) \geq 0, \mathbf{y} \in \mathbb{R}^{D_{\mathrm{X}}}$ of a generalized Poisson process is said to be regular if its integral over the (possibly unbounded) support $\mathcal{R} \subseteq \mathbb{R}^{D_{\mathrm{X}}}$ is finite:

$$\Lambda(\mathcal{R}) = \int_{\mathcal{X}} \lambda(\mathbf{y}) d\mathbf{y} < \infty. \tag{19}$$

A key conclusion from the theory of point processes (Cox & Isham, 1980) is that the probability of event occurrence $\mathbf{X}$ can be formulated as

$$\mathrm{Prob}(\mathbf{X}) = \exp(-\Lambda(\mathcal{R})) \prod_{1 \leq i \leq N} \lambda(\mathbf{x}_i), \tag{20}$$

and the number of points $N = |\mathbf{X}|$ is with respect to a Poisson:

$$\mathrm{Prob}(N = k) = \exp(-\Lambda(\mathcal{R})) \frac{\Lambda(\mathcal{R})^k}{k!}. \tag{21}$$

Based on this fact, we can infer that

$$\mathrm{Prob}(N < \infty) = \sum_{k \geq 0} \mathrm{Prob}(N = k) = \exp(-\Lambda(\mathcal{R})) \sum_{k \geq 0} \frac{\Lambda(\mathcal{R})^k}{k!} = 1, \tag{22}$$

which exactly verifies the second claim in the proposition.

## A.2 ANALYSIS OF THE CONVERGENCE SPEED

The quadratic Wasserstein distance (Ramdas et al., 2017) $\mathcal{W}_2$ between two distributions $\pi_1, \pi_2 :$ $\mathbb{R}^{D_P} \to R$ is defined as

$$\mathcal{W}_2(\pi_1, \pi_2)^2 = \inf_{\lambda \in \Lambda(\pi_1, \pi_2)} \left( \int \|\mathbf{x} - \mathbf{y}\|_2^2 d\lambda(\mathbf{x}, \mathbf{y}) \right),$$

where $\Lambda$ denotes the set of all possible joint distributions, with marginals the same as its arguments.

We first show that metric $\mathcal{W}_2$ enjoys certain decomposability. Suppose that there are any two probability measures $\pi_1, \pi_2$ that shape as a mixture. Formally, we formulate them as

$$\pi_1 = \sum_{1 \leq i \leq n} w_{1,i} \pi_{1,i}, \quad \pi_2 = \sum_{1 \leq j \leq m} w_{2,j} \pi_{2,j}, \tag{23}$$

where $\sum_i w_{1,i} = 1$, $\sum w_{2,j} = 1$, and $\pi_{1,i}, \pi_{2,j}$ are also probability measures. We further suppose that the distance $\mathcal{W}_2(\pi_{1,i}, \pi_{2,j}) < \infty$ is finite for $\forall i, j \in [1, n] \times [1, m]$, then there exists a positive number $c_{i,j}$ that bounds its squared value. By definition we know that there is also a probability measure $\widetilde{\lambda}_{i,j} \in \Gamma(\pi_{1,i}, \pi_{2,j})$ satisfying

$$\int \|\mathbf{x} - \mathbf{y}\|_2^2 d\widetilde{\lambda}_{i,j}(\mathbf{x}, \mathbf{y}) < c_{i,j}. \tag{24}$$

Now, we construct a new probability measure as

$$\widetilde{\lambda} = \sum_{1 \leq i \leq n} \sum_{1 \leq j \leq m} w_{1,i} w_{2,j} \widetilde{\lambda}_{i,j}. \tag{25}$$

It is easy to verify that this new measure $\widetilde{\lambda}$ belong to $\Lambda(\pi_1, \pi_2)$ as

$$\int \widetilde{\lambda}(\mathbf{x}, \mathbf{y}) d\mathbf{x} = \int \left( \sum_{1 \leq i \leq n} \sum_{1 \leq j \leq m} w_{1,i} w_{2,j} \widetilde{\lambda}_{i,j}(\mathbf{x}, \mathbf{y}) \right) d\mathbf{x}$$

$$= \sum_{1 \leq i \leq n} \sum_{1 \leq j \leq m} w_{1,i} w_{2,j} \left( \int \widetilde{\lambda}_{i,j}(\mathbf{x}, \mathbf{y}) d\mathbf{x} \right) = \sum_{1 \leq i \leq n} \sum_{1 \leq j \leq m} w_{1,i} w_{2,j} \pi_{2,j}(\mathbf{y}) \tag{26}$$

$$= \left( \sum_{1 \leq i \leq n} w_{1,i} \right) \left( \sum_{1 \leq j \leq m} w_{2,j} \pi_{2,j}(\mathbf{y}) \right) = 1 \cdot \pi_2(\mathbf{y}) = \pi_2(\mathbf{y}).$$

Symmetrically, we can also prove that $\int \widetilde{\lambda}(\mathbf{x}, \mathbf{y}) d\mathbf{y} = \pi_1(\mathbf{x})$ using the same derivation strategy. Based on the definition of Wasserstein metric $\mathcal{W}_2$, we have

$$\mathcal{W}_2(\pi_1, \pi_2)^2 \leq \int \|\mathbf{x} - \mathbf{y}\|_2^2 d\widetilde{\lambda}(\mathbf{x}, \mathbf{y}) =$$

$$\sum_{1 \leq i \leq n} \sum_{1 \leq j \leq m} w_{1,i} w_{2,j} \int \|\mathbf{x} - \mathbf{y}\|_2^2 d\widetilde{\lambda}_{i,j}(\mathbf{x}, \mathbf{y}) \leq \sum_{1 \leq i \leq n} \sum_{1 \leq j \leq m} w_{1,i} w_{2,j} c_{i,j}. \tag{27}$$

Now, we let the constant $c_{i,j}$ go to its limit: $c_{i,j} \to \mathcal{W}_2(\pi_{1,i}, \pi_{2,j})^2$, leading to

$$\mathcal{W}_2(\pi_1, \pi_2)^2 \leq \sum_{1 \leq i \leq n} \sum_{1 \leq j \leq m} w_{1,i} w_{2,j} \mathcal{W}_2(\pi_{1,i}, \pi_{2,j})^2. \tag{28}$$

Then, we aim to study the Wasserstein distance $\mathcal{W}_2$ between Gaussian $\pi_1 = \int \mathcal{G}(\mathbf{z}_1, \gamma^2 \mathbf{I})$ and delta distribution $\pi_2 = \int \delta_{\mathbf{z}_2}$. Here we abuse the symbol $\int$ to denote an integral-like operator: mapping a density function to its measure. Note that the latter is only supported on a single point: $\mathbf{z}_2$. Therefore, there is in fact only one trivial joint measure: $\lambda = \int \mathcal{G}(\mathbf{z}_1, \gamma^2 \mathbf{I}) \cdot \int \delta_{\mathbf{z}_2}$ in space $\Lambda(\cdot)$. In this sense, we have

$$\mathcal{W}_2(\mathcal{G}(\mathbf{z}_1, \gamma^2 \mathbf{I}), \delta_{\mathbf{z}_2})^2 = \int_{\mathbf{x}} \int_{\mathbf{y}} \|\mathbf{x} - \mathbf{y}\|_2^2 \mathcal{G}(\mathbf{x}; \mathbf{z}_1, \gamma^2 \mathbf{I}) \delta_{\mathbf{z}_2}(\mathbf{y}) d\mathbf{x} d\mathbf{y}$$

$$= \int_{\mathbf{x}} \mathcal{G}(\mathbf{x}; \mathbf{z}_1, \gamma^2 \mathbf{I}) \left( \int_{\mathbf{y}} \|\mathbf{x} - \mathbf{y}\|_2^2 \delta_{\mathbf{z}_2}(\mathbf{y}) d\mathbf{y} \right) d\mathbf{x} = \int_{\mathbf{x}} \mathcal{G}(\mathbf{x}; \mathbf{z}_1, \gamma^2 \mathbf{I}) \|\mathbf{x} - \mathbf{z}_2\|_2^2 d\mathbf{x}$$

$$= \mathbb{E}_{\mathbf{x} \sim \mathcal{G}(\mathbf{z}_1, \gamma^2 \mathbf{I})} \left[ \|\mathbf{x} - \mathbf{z}_2\|_2^2 \right] = \mathbb{E}_{\mathbf{x} \sim \mathcal{G}(\mathbf{0}, \mathbf{I})} \left[ \|\gamma \mathbf{x} + \mathbf{z}_1 - \mathbf{z}_2\|_2^2 \right] \tag{29}$$

$$= \gamma^2 \mathbb{E}_{\mathbf{x} \sim \mathcal{G}(\mathbf{0}, \mathbf{I})} \left[ \|\mathbf{x}\|_2^2 \right] + 2\gamma \mathbb{E}_{\mathbf{x} \sim \mathcal{G}(\mathbf{0}, \mathbf{I})} \left[ (\mathbf{z}_1 - \mathbf{z}_2)^\top \mathbf{x} \right] + \|\mathbf{z}_1 - \mathbf{z}_2\|_2^2$$

$$= \gamma^2 D_P + \|\mathbf{z}_1 - \mathbf{z}_2\|_2^2.$$

Combining the above two key conclusions, we have

$$
\begin{aligned}
\mathcal{W}_2(f_{\mathbf{X},\sigma(\epsilon)}, f_{\mathbf{X}})^2 &\leq \sum_{1\leq i,j\leq N} \frac{1}{N^2} \mathcal{W}_2(\mathcal{G}(\mathbf{x}_i, \sigma_i(\epsilon)^2\mathbf{I}), \delta_{\mathbf{x}_j})^2 \\
&= \frac{1}{N^2} \sum_{1\leq i,j\leq N} \Big(\sigma_i(\epsilon)^2 D_{\mathrm{P}} + \|\mathbf{x}_i - \mathbf{x}_j\|_2^2\Big) \\
&\leq \frac{1}{N^2} \sum_{1\leq i,j\leq N} \Big(\epsilon^2 D_{\mathrm{P}} \min_{k\neq i} \|\mathbf{x}_i - \mathbf{x}_k\|_2^2 + \|\mathbf{x}_i - \mathbf{x}_j\|_2^2\Big) \\
&\leq \frac{1}{N^2} \sum_{1\leq i,j\leq N} \rho^2(\epsilon^2 D_{\mathrm{P}} + 1) = \rho^2(\epsilon^2 D_{\mathrm{P}} + 1).
\end{aligned}
\tag{30}
$$

We can see that the outcome in this derivation has an undesired term $\rho^2$, resulting in a too loose upper bound. To address this problem, we consider a different strategy in this stage: The essence of Wasserstein distance $\mathcal{W}_2$ is to measure the minimum cost of transforming the probability density from one to another. Therefore, any transform plan that we can propose must incur a cost that is not smaller than the distance.

In this spirit, a transform plan $\widetilde{\pi}$ is to move the whole probability mass (i.e., 1) of every Gaussian $\mathcal{G}(\mathbf{x}_i, \sigma_i(\epsilon)^2\mathbf{I})$ in mixture representation $f_{\mathbf{X},\sigma(\epsilon)}$ to the Dirac $\delta_{\mathbf{x}_i}$ in delta representation $f_{\mathbf{X}}$ that has the same center:

$$
\pi(\mathcal{X}, \mathbf{y} = \mathbf{x}_i) = \frac{1}{N} \int \Big[\mathcal{G}(\mathbf{x}_i, \sigma_i(\epsilon)^2\mathbf{I})\Big](\mathcal{X}), \quad \pi(\mathcal{X}, \mathbf{y} \notin \mathbf{X}) = 0,
\tag{31}
$$

where $\mathcal{X}$ is a Borel set in $\mathbb{R}^{D_{\mathrm{P}}}$. Considering the previous conclusion, we have

$$
\begin{aligned}
\mathcal{W}_2(f_{\mathbf{X},\sigma(\epsilon)}, f_{\mathbf{X}})^2 &\leq \int \|\mathbf{x} - \mathbf{y}\|_2^2 d\widetilde{\lambda}(\mathbf{x}, \mathbf{y}) \\
&= \int_{\mathbf{x}} \int_{\mathbf{y}} \|\mathbf{x} - \mathbf{y}\|_2^2 \Big(\frac{1}{N} \sum_{1\leq i\leq N} \mathcal{G}(\mathbf{x}; \mathbf{x}_i, \sigma_i(\epsilon)^2\mathbf{I})\delta_{\mathbf{x}_i}(\mathbf{y})\Big) d\mathbf{x} d\mathbf{y} \\
&= \frac{1}{N} \sum_{1\leq i\leq N} \int_{\mathbf{x}} \int_{\mathbf{y}} \|\mathbf{x} - \mathbf{y}\|_2^2 \mathcal{G}(\mathbf{x}; \mathbf{x}_i, \sigma_i(\epsilon)^2\mathbf{I})\delta_{\mathbf{x}_i}(\mathbf{y}) d\mathbf{x} d\mathbf{y} \\
&= \frac{1}{N} \sum_{1\leq i\leq N} \mathcal{W}_2(\mathcal{G}(\mathbf{x}_i, \sigma_i(\epsilon)^2\mathbf{I}), \delta_{\mathbf{x}_i})^2 \\
&= \frac{1}{N} \sum_{1\leq i\leq N} \Big(\sigma_i(\epsilon)^2 D_{\mathrm{P}} + \|\mathbf{x}_i - \mathbf{x}_i\|_2^2\Big) \\
&= \frac{1}{N} \sum_{1\leq i\leq N} \epsilon^2 \ln\Big(1 + D_{\mathrm{P}} \min_{j\neq i} \|\mathbf{x}_i - \mathbf{x}_j\|_2^2\Big)^2 \leq \epsilon^2 D_{\mathrm{P}}(\ln(1+\rho))^2.
\end{aligned}
\tag{32}
$$

Therefore, we get the conclusion: $\mathcal{W}_2(f_{\mathbf{X},\sigma(\epsilon)}, f_{\mathbf{X}}) = \mathcal{O}(\epsilon \ln \rho \sqrt{D_{\mathrm{P}}})$.

## B  PROOF: REGULARITY OF THE FUNCTION SPACE

For the first part, we only have to prove that the mixture representation $f_{\mathbf{X},\sigma(\epsilon)}$ is always square-integrable. Because if this is the case, an element that is in the space $\mathcal{F}_{\mathrm{mix}}$ also belongs to the $L^2$ space $\mathcal{L}^2(\mathbb{R}^{D_{\mathbf{X}}})$, indicating $\mathcal{F}_{\mathrm{mix}} \subseteq \mathcal{L}^2(\mathbb{R}^{D_{\mathbf{X}}})$. Specifically, we have

$$
\begin{aligned}
\|f_{\mathbf{X},\sigma(\epsilon)}\|_{\mathcal{L}^2}^2 &= \int f_{\mathbf{X},\sigma(\epsilon)}(\mathbf{y})^2 d\mathbf{y} = \int \Big(\sum_{1\leq i\leq N} \frac{1}{N} \mathcal{G}(\mathbf{y}; \mathbf{x}_i, \sigma_i(\epsilon)^2\mathbf{I})\Big)^2 d\mathbf{y} \\
&= \int \Big(\frac{1}{N^2} \sum_{1\leq i,j\leq N} \mathcal{G}(\mathbf{y}; \mathbf{x}_i, \sigma_i(\epsilon)^2\mathbf{I})\mathcal{G}(\mathbf{y}; \mathbf{x}_j, \sigma_j(\epsilon)^2\mathbf{I})\Big) d\mathbf{y} \\
&= \frac{1}{N^2} \sum_{1\leq i,j\leq N} \Big(\int \mathcal{G}(\mathbf{y}; \mathbf{x}_i, \sigma_i(\epsilon)^2\mathbf{I})\mathcal{G}(\mathbf{y}; \mathbf{x}_j, \sigma_j(\epsilon)^2\mathbf{I}) d\mathbf{y}\Big).
\end{aligned}
\tag{33}
$$

It is a known fact (Ahrendt, 2005) that the Gaussian product can be reshaped as

$$\mathcal{G}(\cdot)\mathcal{G}(\cdot) = \mathcal{G}\Big(\mathbf{x}_i; \mathbf{x}_j, (\sigma_i(\epsilon)^2 + \sigma_j(\epsilon)^2)\mathbf{I}\Big)\mathcal{G}\Big(\mathbf{y}; \frac{\sigma_j(\epsilon)^2\mathbf{x}_i + \sigma_i(\epsilon)^2\mathbf{x}_j}{\sigma_i(\epsilon)^2 + \sigma_j(\epsilon)^2}, \frac{\sigma_i(\epsilon)^2\sigma_j(\epsilon)^2}{\sigma_i(\epsilon)^2 + \sigma_j(\epsilon)^2}\mathbf{I}\Big). \quad (34)$$

With this fact, the function form can be simplified as

$$\|f_{\mathbf{X},\sigma(\epsilon)}\|_{\mathcal{L}^2}^2 = \frac{1}{N^2}\sum_{1\leq i,j\leq N}\Big(\int \mathcal{G}(\mathbf{x}_i; \mathbf{x}_j, (\sigma_i(\epsilon)^2 + \sigma_j(\epsilon)^2)\mathbf{I})\mathcal{G}(\mathbf{y};\cdot)d\mathbf{y}\Big)$$

$$= \frac{1}{N^2}\sum_{1\leq i,j\leq N}\mathcal{G}(\mathbf{x}_i; \mathbf{x}_j, (\sigma_i(\epsilon)^2 + \sigma_j(\epsilon)^2)\mathbf{I})\Big(\int \mathcal{G}(\mathbf{y};\cdot)d\mathbf{y}\Big) \quad (35)$$

$$= \frac{1}{N^2}\sum_{1\leq i,j\leq N}\mathcal{G}(\mathbf{x}_i; \mathbf{x}_j, (\sigma_i(\epsilon)^2 + \sigma_j(\epsilon)^2)\mathbf{I}),$$

which is finite because $\sigma_i(\epsilon)^2 + \sigma_j(\epsilon)^2 > 0$. Therefore, the mixture representation is square-integrable $\|f_{\mathbf{X},\sigma(\epsilon)}\|_{\mathcal{L}^2}^2 < \infty$, which proves the claim.

For the second part, it is sufficient to show that the square integral of the delta representation $f_{\mathbf{X}}$ is infinite. Similar to the above derivation, we first decompose the squared norm $\|f_{\mathbf{X}}\|_{\mathcal{L}^2}^2$ as

$$\int\Big(\sum_{1\leq i\leq N}\frac{1}{N}\delta_{\mathbf{x}_i}(\mathbf{y})\Big)^2 d\mathbf{y} = \int\Big(\frac{1}{N^2}\sum_{1\leq i,j\leq N}\delta_{\mathbf{x}_i}(\mathbf{y})\delta_{\mathbf{x}_j}(\mathbf{y})\Big)d\mathbf{y}$$

$$= \frac{1}{N^2}\sum_{1\leq i,j\leq N}\Big(\int \delta_{\mathbf{x}_i}(\mathbf{y})\delta_{\mathbf{x}_j}(\mathbf{y})d\mathbf{y}\Big) = \frac{1}{N^2}\sum_{1\leq i,j\leq N}\delta_{\mathbf{0}}(\mathbf{x}_i - \mathbf{x}_j) \quad (36)$$

$$= \frac{1}{N^2}\Big(N\delta_{\mathbf{0}}(\mathbf{0}) + \sum_{i\neq j}\delta_{\mathbf{0}}(\mathbf{x}_i - \mathbf{x}_j)\Big) = \frac{1}{N}\delta_{\mathbf{0}}(\mathbf{0}) = \infty.$$

where $\mathbf{0}$ represents a vector full of zero. This result verifies the claim, and indicate that $L^2$ space $\mathcal{L}^2(\mathbb{R}^{D_{\mathbf{X}}})$ is not able to support the probability measure $\mu_{\mathcal{F}_{\text{delta}}}$.

## C  WARM-UP EFFECT OF LANGEVIN DYNAMICS

Langevin dynamics (Coffey & Kalmykov, 2012) is a type of Markov chain Monte Carlo (MCMC) (Andrieu et al., 2003) that can convert any initial continuous distribution to a desired one. Specifically, one usually sets the initial distribution at time step $s = 1$ as a very simple distribution (e.g., Gaussian and uniform) that is easy to sample from, and the dynamics will incrementally update the distribution in terms of the score function (i.e., some statistical information about the desired distribution), such that it will converge to the desired one as $s \to \infty$.

To be more rigorous, suppose that one would like to sample from a continuous distribution $\pi_{\text{tgt}}$ and it is easy to sample from some distribution $\pi_{\text{src}}$, then let us see the following Langevin dynamics

$$\mathbf{z}^{(s+1)} = \mathbf{z}^{(s)} + \beta\nabla\ln\pi_{\text{tgt}}(\mathbf{z}^{(s)}) + \sqrt{2\beta}\mathbf{w}^{(s)}, \mathbf{z}^{(1)} \sim \pi_{\text{src}}, \quad (37)$$

where $\mathbf{w}^{(s)}$ is a noise sampled from standard Gaussian $\mathcal{N}(\mathbf{0}, \mathbf{I})$ and $\{\mathbf{z}^{(s)}\}_{s\in\mathbb{Z}^+}$ forms a trajectory that records how an initial particle $\mathbf{z}^{(1)}$ evolves over time. We denote the marginal distribution of particle $\mathbf{z}^{(s)}$ at time step $s$ as $\pi_s$. Trivially, we have $\pi_1 = \pi_{\text{src}}$, and importantly, the following limit

$$\lim_{s\to\infty}\pi_s = \pi_{\text{tgt}}. \quad (38)$$

holds. This type of convergence is proved to be of an exponential speed (Xu et al., 2018), which also runs very fast in practice. Interestingly, this technique can be applied regardless of the type of initial distribution $\pi_0$.

In the framework of *unordered flow*, we apply the Langevin dynamics to warm up the uniformly initialized particles $\mathbf{Y}^{(0)}$. As we can anticipate from the above guide, the empirical distribution of tuned particle set $\mathbf{Y}^{(S_{\text{lgvin}})}$ should be consistent with the mixture representation $\widehat{f}_{\star,\sigma(\epsilon)}$. Therefore, while some particles might be initially located at low-density areas, they will be attracted to high-density regions by the Langevin dynamics. This warm-up trick is effective to make gradient-based search robust to *noisy peaks*, such that the quality of final outcome $\widehat{\mathbf{X}}$ is less likely to be affected by the constant $\epsilon$ and neural network approximation errors.

# D  IMPLEMENTATION OF UNORDERED FLOW

In this part, we first present the neural network parameterization of our *unordered flow* model $\mathbf{u}_{\theta,t}$, which is not trivial as it involves function-valued mapping. Then, we detail the training and inference procedures of the model.

## D.1  PARAMETERIZATION OF THE FUNCTION-VALUED MODEL

Deep neural networks are typically designed to approximate vector-valued function $g : \mathbb{R}^{D_X} \to \mathbb{R}^{D_Y}$, where $D_X, D_Y$ are respectively the dimensions of source and target spaces. A typical formulation is as follows:

$$\mathbf{y} \approx g(\mathbf{x}) = (\mathbf{W}_L \circ \sigma_L \circ \mathbf{W}_{L-1} \circ \sigma_{L-1} \circ \cdots \circ \sigma_2 \circ \mathbf{W}_1)(\mathbf{x}), \tag{39}$$

where $\circ$ means the function composition, $L$ is the number of layers, $\sigma_i, i \in [2, L]$ is some activation function, and $\mathbf{W}_i, i \in [1, L]$ is a learnable matrix. Since our model $\mathbf{u}_{\theta,t} : \mathcal{H}_{L2} \to \mathcal{H}_{L2}$ is a mapping between two Hilbert spaces, with its input and output being infinite-dimensional, the conventionally defined neural networks are inapplicable.

A mature technique: neural operator (Kovachki et al., 2023; Anandkumar et al., 2019), in the literature can be applied in our setting, which models the mapping from one function to the other. Its general formulation is as

$$h_Y[\mathbf{z}] = v(h_X)[\mathbf{z}] = (Q \circ K_L \circ \sigma_L \circ K_{L-1} \circ \sigma_{L-1} \circ \cdots \circ \sigma_2 \circ K_1 \circ P)h_X[\mathbf{z}], \tag{40}$$

where $L$ is the number of operator compositions, $\sigma_i, i \in [2, L]$ is still some activation function, and $K_i, i \in [1, L]$ is a learnable linear operator, with matrices $P, Q$ being element-wise operations that respectively increase and decrease the channel dimension. To make the notation simpler, let us suppose that $P, Q$ maintain the channel dimension. In that case, each linear operator $K_i$ can be expressed as a kernel form:

$$K_i(h)[\mathbf{z}] = \int k_i(\mathbf{z}, \mathbf{y}) h(\mathbf{y}) d\mathbf{y}, \tag{41}$$

where $k_i : \mathbb{R}^{D_z} \times \mathbb{R}^{D_z} \to \mathbb{R}$ is a learnable kernel function that can be parameterized by conventional vector-valued neural networks. In practice, the integral on the right hand side needs to be approximated by discretization.

Specifically, we adopt the Fourier neural operator (Li et al., 2021) as the backbone to implement *unordered flow* $\mathbf{u}_{\theta,t}$. The basic idea is that, if the kernel $k_i$ is position-invariant (e.g., Gaussian), then the linear operator $K_i$ can be interpreted as some type of convolution. Specifically, it will take an expression as

$$K_i(h)[\mathbf{z}] = \int k_i(\mathbf{z} - \mathbf{y}) h(\mathbf{y}) d\mathbf{y} = (k_i \otimes h)[\mathbf{z}]. \tag{42}$$

In terms of Fourier transform $\mathcal{F}$, the leftmost term can be further reshaped as

$$K_i(h)[\mathbf{z}] = \mathcal{F}^{-1}(\mathcal{F}(k_i)\mathcal{F}(h))[\mathbf{z}] = \mathcal{F}^{-1}(\mathbf{w}_i \mathcal{F}(h))[\mathbf{z}], \tag{43}$$

where $w_i : \mathbb{R}^{D_z} \to \mathbb{C}$ is a learnable neural network and $\star^{-1}$ means to take the operator inverse. This type of neural operator gets around the intractable integral, with many efficient implementations for Fourier transform.

## D.2  TRAINING AND INFERENCE PROCEDURES

The training procedure of our *unordered flow* model is shown in Algorithm 2. Compared with vanilla flow matching, the major difference is that the conditional sample drawn from $\mu_1$ is a mixture-like function $f_{\mathbf{X},\sigma(\epsilon)}$, and the noise function $h$ comes from a Gaussian measure. In our experiments, we implement this type of non-Lebesgue measure through a zero-mean Gaussian process (Seeger, 2004) parametrized by a Matérn kernel. Another key point is that the computation of loss $\widehat{\mathcal{J}}_H$ involves discretization, sampling a number of points that locate in $\mathbb{R}^{D_X}$. Obviously, we need to sample the most informative points such that our model training is sample-efficient. The easiest way to achieve this goal is to regard the function $f_{\mathbf{X},\sigma(\epsilon)}$ as some probability density (as mentioned in Sec. 3.3), and

---

**Algorithm 2:** Training Algorithm

---

**Input:** Dataset $\mathcal{X}$, training iterations $L$, constant $\epsilon$, covariance operator $\Gamma$, learning rate $\eta$
**Output:** Trained model $\mathbf{u}_{\theta,t}$
**for** $i \in [1, 2, \cdots, L]$ **do**
  Randomly sample an instance of unordered data: $\mathbf{X}$, from the training set $\mathcal{X}$
  Represent the data $\mathbf{X}$ as a mixture representation $f_{\mathbf{X},\sigma(\epsilon)}$ (i.e., Eq. (5))
  Randomly sample a time step $t$ from uniform distribution $\mathcal{U}\{0, 1\}$
  Randomly sample a Gaussian noise $h$ from the initial probability measure $\eta_0 = \mathcal{G}(\mathbf{0}, \Gamma)$
  Compute the intermediate variable $h_{\mathrm{mid}} = \varphi_t(h \mid f_{\mathbf{X},\sigma(\epsilon)})$ (i.e., Eq 8)
  Compute the conditional vector field $\mathbf{u}_t(h \mid f_{\mathbf{X},\sigma(\epsilon)})$ (i.e., Eq. (9))
  Compute the MSE $\widehat{\mathcal{J}}_{\mathrm{H}} = \|\mathbf{u}_{\theta,t}(h_{\mathrm{mid}}) - \mathbf{u}_t(h_{\mathrm{mid}} \mid f_{\mathbf{X},\sigma(\epsilon)})\|_{\mathrm{L2}}^2$ (i.e., Eq. (10))
  Perform gradient descent as $\theta = \theta - \eta\nabla_\theta \widehat{\mathcal{J}}_{\mathrm{H}}$

---

**Algorithm 3:** Sampling Algorithm

---

**Input:** Trained model $\mathbf{u}_{\theta,t}$, bounded region $\mathcal{X}$, warm-up steps $S_{\mathrm{lgvin}}$, search steps $S_{\mathrm{grad}}$
**Output:** Unorded point set $\widehat{\mathbf{X}}$
Sample a function representation $\widehat{f}_{\star,\sigma(\epsilon)}$ from the model $\mathbf{u}_{\theta,t}$
Sample a particle set $\mathbf{Y}^{(1)}$ from region $\mathcal{X}$
**for** $s = 1, 2, \cdots S_{\mathrm{lgvin}} - 1$ **do**
  Update with Langevin warm-up: $\mathbf{Y}^{(s)} \mapsto \mathbf{Y}^{(s+1)}$, in terms of Eq. (12)
**for** $s = S_{\mathrm{lgvin}}, S_{\mathrm{lgvin}} + 1, \cdots, S_{\mathrm{lgvin}} + S_{\mathrm{grad}} - 1$ **do**
  Update with gradient ascent: $\mathbf{Y}^{(s)} \mapsto \mathbf{Y}^{(s+1)}$, in terms of Eq. (11)
Merge particles into groups and exclude small ones: $\widehat{\mathbf{Y}} = \mathbf{Y}^{(S_{\mathrm{lgvin}}+S_{\mathrm{grad}})} \mapsto \widehat{\mathbf{X}}$

---

perform sampling with respect to it: At every time, we first randomly select a Gaussian component and then draw a sample from the component.

The inference procedure of our model is demonstrated in Algorithm 3. For the step of gradient ascent, an important point is how to compute the gradient of a mixture representation: $\nabla \widehat{f}_{\star,\sigma(\epsilon)}(\mathbf{y})$, which also plays a key role in Langevin warm-up because the score function can be calculated as

$$\nabla \ln \widehat{f}_{\star,\sigma(\epsilon)}(\mathbf{y}) = \nabla \widehat{f}_{\star,\sigma(\epsilon)}(\mathbf{y}) / \widehat{f}_{\star,\sigma(\epsilon)}(\mathbf{y}). \tag{44}$$

Based on Eq. (7), the representation $\widehat{f}_{\star,\sigma(\epsilon)}$ is compute as below:

$$\widehat{f}_{\star,\sigma(\epsilon)} = \varphi_1(h) = \int_0^1 \mathbf{u}_{\theta,t}(\varphi_t(h))dt \approx \frac{1}{|\mathcal{T}|}\sum_{t\in\mathcal{T}} \mathbf{u}_{\theta,t}(\varphi_t(h)), \tag{45}$$

where $\mathcal{T}$ is a set of time steps that are regularly distributed in interval $[0, 1]$ and $h$ denotes a pure noise sampled from some Gaussian measure $\mathcal{G}(\mathbf{0}, \Gamma)$. By applying the gradient operator $\nabla$, we have

$$\nabla_\mathbf{y}\left(\widehat{f}_{\star,\sigma(\epsilon)}\right)[\mathbf{y}] \approx \frac{1}{|\mathcal{T}|}\sum_{t\in\mathcal{T}} \nabla_\mathbf{y}\left(\mathbf{u}_{\theta,t}(\varphi_t(h))\right)[\mathbf{y}], \tag{46}$$

where the gradient term on the right hand side is obtained through the automatic differentiation mechanism of current deep learning frameworks (e.g., PyTorch (Paszke et al., 2019)). Importantly, the two terms: representation $\widehat{f}_{\star,\sigma(\epsilon)}$ and its gradient $\nabla \widehat{f}_{\star,\sigma(\epsilon)}$, can be jointly computed, following the same generation trajectory: $\{\varphi_t(h)\}_{t\in\mathcal{T}}$.

## E    DISCUSSION ON GRAPH-BASED GENERATIVE MODELS

Graph generation models (Vignac et al., 2023; Bose et al.) are typically not appropriate for set-structured data modeling, though a graph is more complex than a point set. Here is a detailed

| Size Argument $s = s'$ | $s' = 10$ | $s' = 50$ | $s' = 100$ |
|---|---|---|---|
| Mean of Set Size $|\mathbf{X}'|$ | 9.98 | 49.96 | 100.10 |
| Std of Set Size $|\mathbf{X}'|$ | 0.43 | 0.62 | 1.09 |
| Accuracy $\mathbb{E}[\mathbb{1}(|\mathbf{X}'| = s')]$ | 98% | 93% | 91% |

Table 3: The experiment results on the Hawkes dataset, showing that our conditional model can achieve precise cardinality control.

explanation: For graph data $G = (V, E)$, the absence of edge set $E$ means no prior knowledge of node dependencies, so a graph generation model will consider the potential connection between any two nodes $v_1, v_2 \in V$. In this regard, while a graph $G = (V, )$ without "explicit edges" $E$ indeed reduces to an unordered set $X = V$, such a fully-connected graph model cannot characterize some basic point set $X$. For example, Poisson process where set elements $\{x\}_{x \in X}$ are independent random variables sampled in terms of the intensity function $\lambda(x)$, and Wiener process where every element $x$ has a temporal dependent $x'$.

Besides the above gap in principle, the practical implementation of graph models also makes them hard to generate set-like data. For example, the node set $V$ is categorical and its size is fixed in most cases. Notably, many graph-based models (Bose et al.; Yim et al., 2024) inearized the graph $G$, introducing an order to the node set $V$. For example, the protein backbone is arranged as a frame sequence $SE(3)^N$ in Bose et al..

# F  MORE EXPERIMENT DETAILS

Our model and the baselines from our implementations are trained with the Adam optimizer (Kingma & Ba, 2015). The learning rate ranges from $10^{-4}$ to $10^{-3}$, depending on specific architectures, with a dropout ratio set (Srivastava et al., 2014) as 0.1. For the Poisson process, we set its intensity function as an square-exponential mixture:

$$\lambda_{\text{poisson}}(\mathbf{x}) = \mu \sum_{1 \leq i \leq 3} w_i \exp(-\|\mathbf{x} - \mathbf{b}_i\|_2^2). \tag{47}$$

In our proof-of-concept experiment, the parameters $w_i, \mathbf{b}_i, i \in \{1, 2, 3\}$ are set as

$$\mu = 1.0, w_1 = 0.3, w_2 = 0.3, w_3 = 0.4,$$
$$\mathbf{b}_1 = [2.51, 3.12]^\top, \mathbf{b}_2 = [-2.01, -1.12]^\top, \mathbf{b}_3 = [2.51, -2.31]^\top. \tag{48}$$

For the Hawkes process, we adopt a Gaussian kernel to model its self-exciting intensity:

$$\lambda_{\text{hawkes}}(t \mid \mathcal{H}) = \mu + \sum_{s \in \mathcal{H}} \alpha \exp(-\beta(t - s)), \tag{49}$$

where $\mathcal{H}$ denotes the collection of previously occurred events, with $\mu, \alpha, \beta$ respectively set as $0.5, 0.5, 3.0$ in the experiment.

# G  EXTENDED EXPERIMENTS

In this section, we present additional experimental results to supplement Sec. 5 in the main text.

## G.1  SET SIZE CONTROL

In Sec. 5.2, we showed that *unordered flow* can generate point sets that are consistent with the real data in terms of both point locations and set sizes. We highlight that the model can also achieve precise cardinality control through classifier-free guidance (Ho & Salimans, 2021). For implementation, we extend the flow matching model $\mathbf{u}_{\theta,t}(\cdot)$ to accept set size $s$ as a new conditional argument, yielding $\mathbf{u}_{\theta,t}(\cdot, s)$. At the training stage, we pair every point set $\mathbf{X}$ with its cardinality $s = |\mathbf{X}|$ as a conditioning variable for jointly optimizing the new model $\mathbf{u}_{\theta,t}(\cdot, s)$. During inference, we simply fix $s = s'$ to generate the set $\mathbf{X}'$ with desired size $|\mathbf{X}'| = s'$.

| Method | S-WStein | D-MMD |
|---|---|---|
| Our Model w/ Neural Diffusion Process (Dutordoir et al., 2023) | 0.041 | 0.179 |
| Our Model w/ Denoising Diffusion Operator (Lim et al., 2023) | 0.039 | 0.187 |
| Our Model w/ Flow Matching | **0.023** | **0.125** |

Table 4: Comparison experiments between diffusion models and flow matching for function-valued generative modeling on the Earthquakes dataset.

As demonstrated in Table 3, we empirically evaluate this approach on the Hawkes dataset, with each result supported by 100 generated samples. We can see that our guided model achieves exceptional precision in set size control. While minor deviations occur in rare cases, these can be trivially fixed through resampling.

### G.2 EXPLORATION ON DIFFUSION MODELS

Flow matching is the second part of our model, with the first part as set representation $\mathbf{X} \mapsto f_{\mathbf{X},\sigma(\epsilon)}$ and the third one as particle-based inverse transform $f_{\mathbf{X},\sigma(\epsilon)} \mapsto \mathbf{X}$. This second module can alternatively be implemented using a function-valued diffusion model, and we show the the corresponding experiment results in Table 4. Besides the formulation simplicity of flow matching, we can see that it empirically delivers better performance.

