# OpenReview forum: "Flow Matching on Unordered Sets"
_ICLR.cc/2026/Conference — Submitted to ICLR 2026_

### Official Review · Reviewer_SriH · 2025-10-30

**Soundness:** 2
**Presentation:** 3
**Contribution:** 2
**Rating:** 4
**Confidence:** 3

**Summary:**

In this work, the authors present building flow matching on unordered point sets. In particular, a lifting method first converts unordered data into a function representation and a flow matching model is trained on function-valued mapping. The authors further introduce a particle filtering method to generate from the learned function representation. Experiments on synthetic and real-world benchmarks shows the proposed method achieves competitive performance in generating unordered set.

**Strengths:**

1. The paper is well-written and easy to follow.
2. This work investigates flow matching models on unordered sets which is a valuable direction for building more general generative models.
3. On benchmarks shown (e.g., Earthquakes and COVID-19), the proposed method shows competitive performance.

**Weaknesses:**

1. The empirical study is on small scaled dataset. It's unclear how the proposed method works on larger scale.
2. Another line of works that builds generative models on function space [1,2] is also related to the proposed method, but they are not discussed in the paper.

Reference:
[1] Dupont, Emilien, et al. "From data to functa: Your data point is a function and you can treat it like one." arXiv preprint arXiv:2201.12204 (2022).
[2] Du, Yilun, et al. "Learning signal-agnostic manifolds of neural fields." Advances in Neural Information Processing Systems 34 (2021): 8320-8331.

**Questions:**

1. What is the size of datasets in empirical study? It would help better understand the scale of the benchmark.
2. What is the architecture of the flow matching model? And also what is the model size compared to baselines?
3. I think the proposed method can be applied to broader problems like 3D point cloud generation. Did the authors try 3D generative tasks and how the proposed method works?
4. In Eq.5, to adaptive rescale the Gaussian variance, one will need to compute pairwise distances over all points. I'm curious how much compute overhead this leads to, especially for data with large number of points.

---

> ### Author Response · Authors · 2025-12-01
>
> Dear Reviewer SriH,
>
> We thank you for your review.
>
> Questions about related work: Another line of works that builds generative models on function space…
>
> Answer: Functa [1] and GEM [2] both convert each datapoint into a neural field. Specifically, Functa emphasizes creating a unified function-parameter representation for images and NeRFs, building the standard generative architectures in function space, whereas GEM focuses on discovering a low-dimensional manifold of neural fields and generating by sampling latent codes on that manifold. Our method, in contrast, is not about neural-field representation: it specifically addresses unordered point sets, constructing a function representation that is intrinsically permutation-invariant, training a function-valued flow tailored to this set representation.
>
> Questions about data: What is the size of datasets in empirical study… Did the authors try 3D generative tasks and how the proposed method works...
>
> Answer: We followed the real datasets used by previous works, with each containing about tens to hundreds of points per sample. Please refer to Chen et al. (2021) for the details. For example, the number of events per sample in the Earthquake dataset ranges between 18 to 543. The implementation of our method that is based on the neural operators worked well in these benchmarks, as shown in our paper, but its memory usage and computational overhead will grow much in terms of larger dimensions. The main limit is the computational device (e.g., GPU), as other works in scientific machine learning. From another perspective, 1D/2D data should already cover many real scenarios (e.g., geological data), and we remain the exploration of 3D for future work.
>
> Questions about the method: …one will need to compute pairwise distances over all points…
>
> Answer: We do not need to compute all pairwise distances. We only need to find the nearest neighbors for each point, which can be done efficiently using data structures like KD-trees.

---

### Official Review · Reviewer_QqoW · 2025-10-31

**Soundness:** 3
**Presentation:** 3
**Contribution:** 2
**Rating:** 6
**Confidence:** 3

**Summary:**

This paper presents the unordered flow that converts unordered data into the function representation with adaptive variances and learn it through function-valued flow matching. To return back to the unordered set from the function representation, the paper uses a particle filtering method that randomly initializes the particles with Langevin dynamics and then performs gradient ascent on the particle updates. Moreover, the paper proposes de-duplication procedure as noise filtering to handle redundancy. Finally, the paper experiments on multiple real-world datasets to show the effectiveness of the proposed method on generating set-structured data.

**Strengths:**

•	The paper is well-written and clearly-organized.

•	The details of the method are considered in depth. The adaptive variance in function representation, the de-duplication as noise filtering are designations that improve the sample quality.

•	The experiments and the ablation studies show the effect of the adaptive variance, the Langevin warm up and the noisy peak filtering.

**Weaknesses:**

•	The sample efficiency of the proposed method seems to be low due to the redundancy during the conversion between the unordered set and the function representation.

•	It would be beneficial to compare the proposed method with some recent flow matching method on point cloud generation like [1], since the main design is to use function-valued flow matching.

•	[1] Lan, Y., Zhou, S., Lyu, Z., Hong, F., Yang, S., Dai, B., ... & Loy, C. C. (2024). GaussianAnything: Interactive Point Cloud Flow Matching For 3D Object Generation. arXiv preprint arXiv:2411.08033.

**Questions:**

•	Why exactly does the Langevin warm-up “effective to make gradient-based search robust to noisy peaks”? It seems that it is still possible that the sampling being trapped in the local maximums after the warm-up, since the target distribution still has spurious modes. Can this be explained in mathematical details?

•	In the second step of the de-duplication procedure, how to determine “small in size” quantitatively?

•	What is the relationship between the proposed method and other flow matching method on point cloud generation? Is the conversion between the unordered set and the function representation sample efficient?

---

> ### Author Response · Authors · 2025-12-01
>
> Dear Reviewer QqoW,
>
> We thank you for your review.
>
> Question-1: …It seems that it is still possible that the sampling being trapped in the local maximums after the warm-up…
>
> Answer: Langevin dynamics not only attracts sampling points toward density peaks but also allocates more points to higher peaks. In other words, 'spurious modes' correspond to low peaks and, as a result, very few points end up near them. The significant difference in set sizes makes it easy to distinguish true modes from spurious ones.
>
> Question-2: In the second step of the de-duplication procedure, how to determine “small in size” quantitatively?
>
> Answer: As mentioned in our initial response, points near spurious modes are far fewer than those near true modes. Therefore, for $N$ sampling points, we set a small threshold $x$ (e.g., 5%), and if a mode contains fewer than $N * x%$, we identified it as spurious. In practice, we found that neural networks learn effectively from the data and generate almost no spurious modes. The de-duplication procedure is included primarily for robustness and performs well in those rare cases.
>
> Question-3: What is the relationship between the proposed method and other flow matching method on point cloud generation…
>
> Answer: GaussianAnything (Lan et al. (2024)) is a 3D-focused generative model that builds a point-cloud-structured latent space using a VAE trained on multi-view RGB-D-Normal renderings, and then applies a two-stage flow-matching process to first generate geometry and then generate point-wise texture features. Its decoder produces surfel Gaussians, which support high-fidelity rendering and enable intuitive 3D editing because geometry and texture are naturally disentangled. It also supports multimodal conditioning from text, images, or point clouds. In contrast, our method is a general framework for modeling unordered point sets. Instead of operating directly in point-cloud space, it maps the point set to a functional representation, whose inverse requires additional computation at inference time. Consequently, GaussianAnything is specialized for editable, high-fidelity 3D object generation, while our work offers a broad, principled approach for generating unordered sets.

---

### Official Review · Reviewer_hkEG · 2025-11-01

**Soundness:** 3
**Presentation:** 3
**Contribution:** 2
**Rating:** 4
**Confidence:** 3

**Summary:**

The paper proposes a generative modelling framework for unordered sets (e.g., point‐sets) via, mixture function representation, flow matching on function space, and an inverse transform. The mixture representation ensures the permutation invariance. The generative pipeline include: a flow-based function generation, and an inverse transform to map the function to a discrete set via a two-stage process (warm‐up with Langevin dynamics on particles + gradient‐based refinement).

**Strengths:**

1. The paper has clear motivation. The unordered nature of sets is a real and non‐trivial challenge in generative modelling

2. The proposed approach is novel. By mapping the point sets into mixture function allows permutation invariance and naturally leads to a flow model in function space.

3. They showed performance gains on multiple real-world datasets. .

**Weaknesses:**

1. It is not very clear that the flow model is guaranteed to generate the desired mixture function, with $N$ mixtures. For example, it may introduce spurious modes.

2. I think the description on the inverse mapping to recover the points from the generated mixture representation using the trained model $u_{\theta, t}$ is very heuristic. There is no recovery guarantee. Identifying the mixture parameters itself is not trivial.

3. The expeirmental results is limited (lack higher dimensional case) and some parts are not very clear. For example, it is not very clear why the generated function in Fig. 2(b) is consistent with Fig. 2(a) training data. They appear to have different set sizes.

**Questions:**

1. For general audience, it might be a bit difficult to build a mental picture of the distribution of the functions drawn from $\eta_0$. Could the authors provide some examples, and provide some intuition on why it is a good base distribution?

2. Does the inversion procedure produce exactly valid sets?

3. How does the performance of the proposed algorithm scale with the data $D_X$ increases? In the synthetic experiments, the authors only showed point clouds in 2D. What happens if you consider 3D point clouds, or even higher-dimensional point clouds?

4. How does the performance scale with the number of points $N$?

5. Could the authors report computational cost (training time, inference time) vs other methods; include scaling experiments for varying set size $N$.

---

> ### Author Response · Authors · 2025-12-01
>
> Dear Reviewer hkEG,
>
> We thank you for your review.
>
> Question-1: …it might be a bit difficult to build a mental picture of the distribution of the functions drawn from…
>
> Answer: While the Gaussian measure over function space may feel abstract, its samples can be visualized as smooth, randomly fluctuating curves whose variability is controlled by the covariance kernel. These functions resemble natural trajectories commonly encountered in time-series and physical processes, making the distribution intuitive and flexible. Using a Gaussian measure as the base distribution in flow matching is effective because it provides: 1) a well-understood, analytically tractable prior; 2) smoothness and regularity that stabilize training; 3) sufficient expressive support to connect to a wide range of target function distributions.
>
> Question-2: Does the inversion procedure produce exactly valid sets?
>
> Answer: Please refer to Sec. 3.3 of our paper: the trained model might indeed produce spurious modes that have low function values, so we proposed to adopt Langevin dynamics to guide sampling points towards the true modes that have high function values. The results in Fig. 3 of our paper showed that this method worked well in practice. Alternatively, note that we can also easily identify spurious modes in terms of their density values (i.e., just setting a threshold and filtering them), though this way seems less elegant and needs extra parameter selection on the validation set. We also would like to emphasize that, empirically, the trained model behaves very well at test time, and generates very few spurious modes. Our discussion in the paper regarding this part was just included to handle rare ill-defined cases.
>
> Other questions about scaling: How does the performance of the proposed algorithm scale with the data D_X… How does the performance scale with the number of points N… computational cost...
>
> Answer: The synthetic data and real data we followed from previous works are either 1D or 2D, containing tens to hundreds of points. The implementation of our method that is based on the neural operators worked well in these benchmarks, as shown in our paper, but its memory usage and computational overhead will grow much in terms of larger dimension and number of points. The main limit is the computational device (e.g., GPU), as in other works in scientific machine learning. From another perspective, 1D/2D data should already cover many real scenarios  (e.g., geological data), and we remain the exploration of data scaling (e.g., other types of implementation) for future work.

---

### Official Review · Reviewer_u45W · 2025-11-03

**Soundness:** 3
**Presentation:** 3
**Contribution:** 2
**Rating:** 2
**Confidence:** 3

**Summary:**

This paper extends the flow matching model to unordered point sets. It presents the problem of unordered data generation as generating a function that defines a set of Gaussian distributions centered at the observed points with adaptive variances. The authors then train a functional flow matching to generate the function and use particle filtering to zero in on the points. With experiments on the benchmark unordered point sets, they show superior performance of their method compared to recent baselines.

**Strengths:**

1. The method's presentation was clear. I appreciated the clarity with which each theoretical result was explained
2. The experiments do show a higher distributional similarity compared to the baselines

**Weaknesses:**

Major:

1. The approach appears to be very similar to that of Bilos et al., where the authors used noise from a stochastic process (a Gaussian process) to add noise to the temporal point sets. I acknowledge that the current method (unordered flow) extends this to any point set. However, the experiments in the current paper show temporal point processes only; therefore, without comparing with Bilos et al. (empirically or theoretically), it is difficult to judge the effectiveness of the current approach.

2. The current experiments section only explores unconditional generation. However, the real usefulness of a generative model is the controlled generation. Is there an advantage of this method in conditional generation vs others?

Minor:

Can the author present a quantitative comparison vs the baselines for the proof-of-concept studies?

**Questions:**

Comments:

Page 3 line 144 - the second Gaussian should have $y_j$ as its center.

---

> ### Author Response · Authors · 2025-12-01
>
> Dear Reviewer u45W,
>
> We thank you for your review.
>
> Question-1: The approach appears to be very similar to that of Bilos et al… However, the experiments in the current paper show temporal point processes only…
>
> Answer: Please refer to Fig. 1 and Table 1 of our paper, we conducted experiments not only on temporal point processes but also on spatial point sets. The spatial setting lies outside the application scope of Bilos et al. (2023), making their work fundamentally different from our paper. In essence, time series can be naturally lifted to a function space—as done in Bilos et al. (2023)—whereas this is not true for a spatial process. A major part of our paper focuses on how to represent a point set as a proper function representation so that flow matching can be applied to it.
>
> Question-2: …Is there an advantage of this method in conditional generation vs others?
>
> Answer: The backbone of our model is flow matching, where the velocity prediction model $v$ can be conditioned on extra information $c$. Please refer to Ho et al. (2022) and Lipman et al. (2023), which is a very standard practice. Therefore, our model is capable of conditional generation, just as in the case of other flow and diffusion-based models.
>
> Question-3: …a quantitative comparison vs the baselines for the proof-of-concept studies?
>
> Answer: The proof-of-concept experiments were designed to visually showcase that our model can learn from typical point sets and generate mixture representations. For quantitative comparisons between our model and baselines, please refer to Table 1 of our paper regarding real benchmark datasets.
>
> Question-4: Page 3 line 144 - the second Gaussian should have y_i as its center.
>
> Answer: Thanks for pointing this out. We will fix it in the revised version.

---

### Meta-Review · Area_Chair_vkCp · 2025-12-15

**Summary:**

Reviewers appreciate that the paper is well-written and that the empirical results show promise against the considered baselines.

The main weaknesses highlighted by the reviewers include:
- [W1] Limited discussion and comparison against related works (u45W, QqoW, SriH)
- [W2] Limited empirical evaluations (small datasets, low dimensions, unconditional generation only, etc.) (u45W, hkEG, SriH)
- [W3] Concerns that the proposed approach is heuristic and not fully justified (hkEG)
- [W4] Questions on computational efficiency (hkEG, SriH)

**Reviewer Concerns:**

- [W1] The authors discuss the specific related works brought up by the reviewer. However, given that this was asked by 3/4 reviewers, it indicates that the paper should be updated more significantly to carefully place its contributions within the existing literature on flow matching, function space models, and point processes. The authors do not state they will update their paper in this way, and so this point i only partially resolved.
- [W2] The authors discuss these points in their rebuttal, but do not provide additional evaluations or claim that they will include these. Thus the point is largely outstanding still.
- [W3] The authors do not address this point and it remains outstanding.
- [W4] While there is some discussion of the computational aspects of the proposed approach, the authors do not provide any concrete results regarding training/inference efficiency, leaving this weakness open.

**Reviewer Scores:**

- u45W may have increased their score slightly (e.g., 4) given the discussion on related work, but their comment regarding unconditional generation was not addressed, and a higher score seems unlikely.
- hkEG asks if the proposed approach can be shown to actually sample from the correct distribution, and comments on limited experimental results and discussion on computational efficiency. Without additional results along these lines, the reviewer seems unlikely to change their score.
- QqoW is concerned about sample efficiency and seems unlikely to change their score without concrete results in this direction.
- SriH asks about larger scale datasets and additional comparisions against related techniques. Without such new results, the reviewer seems unlikely to change their score.

Overall, the reviewers agree that this paper proposes an interesting idea which seems promising, but with limited scope of evaluation. I highly encourage the authors to incorporate the feedback gained from the reviewers and to resubmit their work at a later date.

---

### Decision · Program_Chairs · 2026-01-26

Reject